# RISK-SENSITIVE VARIATIONAL MODEL-BASED POLICY OPTIMIZATION

## ABSTRACT

RL-as-inference casts reinforcement learning (RL) as Bayesian inference in a probabilistic graphical model. While this framework allows efficient variational approximations, it is known that model-based RL-as-inference learns optimistic dynamics and risk-seeking policies that can exhibit catastrophic behavior. By exploiting connections between the variational objective and a well-known risk-sensitive utility function, we adaptively adjust policy risk based on the environment dynamics. Our method, $\beta$-VMBPO, extends the variational model-based policy optimization (VMBPO) algorithm to perform dual descent on risk parameter $\beta$. We provide a thorough theoretical analysis that fills gaps in the theory of model-based RL-as-inference by establishing a generalization of policy improvement, value iteration, and guarantees on policy determinism. Our experiments demonstrate that this risk-sensitive approach yields improvements in both tabular and complex continuous tasks, such as the DeepMind Control Suite.

## 1 INTRODUCTION

Casting reinforcement learning (RL) as probabilistic inference provides a useful formalism to develop novel RL algorithms in complex and continuous domains (Levine, 2018; Todorov, 2008; Toussaint, 2009; Kappen et al., 2012; Rawlik et al., 2012). This so-called *RL-as-inference* framework facilitates a principled solution to the exploration-exploitation trade-off by adapting the policy to posterior uncertainty. It also yields flexible models that can incorporate task-knowledge and provides a rich toolbox for effective inference over quantities of interest (Koller & Friedman, 2009). These properties are particularly valuable when the agent has to reason about partial observability and data collection is expensive, making sample efficiency a priority.

A key difference between classical RL and RL-as-inference is the optimization goal. The classical setting aims to discover a policy that produces trajectories with maximum expected return. Conversely, RL-as-inference searches to maximize the probability of optimal trajectories. Yet, despite its promise it has been observed that this latter objective can produce unwanted risk-seeking behaviour in the learned policy (Levine, 2018). This behaviour arises as the agent tends to act optimistically by disregarding the likelihood of low-probability transitions in favor of achieving high potential return.

To address this risk-seeking behaviour two variational approaches dominate the field of RL-as-inference. The first approach removes the notion of control on the variational dynamics, resulting in model-free algorithms — methods that learn direct mappings from states to actions using only samples. This approach, known as *maximum entropy RL* (MaxEnt RL), constrains the variational approximation of posterior dynamics, resulting in loose bounds on the learning objective (Ziebart et al., 2008; Haarnoja et al., 2017; 2018; Fox et al., 2016). The constraint arises from the assumption that an agent with full control over dynamics will preferentially assume unlikely environment transitions. The benefit of this model-free approach is that it avoids explicit evaluation of the model dynamics, but learning can require many environment interactions as a result. Numerous algorithms arise from this approach including: policy search using expectation maximization (EM) (Dayan & Hinton, 1997; Hachiya et al., 2009; Abdolmaleki et al., 2018b) and variational policy search (Haarnoja et al., 2017; 2018).

The second approach does not place restrictions on the variational dynamics, allowing more flexible approximations with tighter bounds on the learning objective. These methods are model-based

as they learn a predictive model for the posterior dynamics. An example of this approach is the variational model-based policy optimization (VMBPO) (Chow et al., 2021), which optimizes a variational bound with an EM-style approach. Another example of this line of work is "Mismatched No More" (MnM) (Eysenbach et al., 2022), which optimizes a similar objective to VMBPO but modifies its reward function. Although these methods can be sample efficient, the learned dynamics are optimistic and risk-seeking, often producing policies that don't generalize to the real environment.

The development of model-based approaches such as VMBPO has stalled due to a lack of study on the risk-seeking behaviour of learned policies. In this work, we present a comprehensive study of risk sensitivity in the model-based RL-as-inference framework. As a practical approach we present $\beta$-VMBPO, a generalization of VMBPO that adapts policy risk by modulating the allowable divergence between posterior and prior dynamics during learning. This method comes as the result of augmenting the log-likelihood presented in Levine (2018) with a hyperparameter $\beta$ that interpolates between conventional RL and the RL-as-inference objective. We rigorously study the impact that $\beta$ has on learning variational dynamics and policies that generalize to the real environment and propose an automatic tuning method based on Lagrangian optimization. Finally, we compare the performance of our method against other RL-as-inference algorithms such as VMBPO (Chow et al., 2021), model-based policy optimization algorithm (MBPO) (Janner et al., 2019), and soft actor critic (SAC) (Haarnoja et al., 2018). Another reason for the slow pace of development in model-based approaches has been the lack of implementation availability. To alleviate the issue, we provide the first open-source implementation of VMBPO and our generalized $\beta$-VMBPO at the time of writing.

Finally, we provide a thorough analysis that establishes fundamental theoretical properties of the risk sensitive RL-as-Inference framework. In particular, we show that the required Bellman operator is a contraction (Theorem 4.1) and establish a generalization of the policy improvement theorem for this setting (Theorem 4.2). Based on these results we show that there exists an optimal deterministic policy (Theorem 4.3). We conclude by connecting our algorithm to the theory by showing that a deterministic policy is optimal under the $\beta$-VMBPO objective (Theorem 4.4).

## 2 PRELIMINARIES: RL AS PROBABILISTIC INFERENCE

The reinforcement learning framework consists of a Markov decision process (MDP) defined by a tuple $(\mathcal{S}, \mathcal{A}, p, r)$. $\mathcal{S}$ and $\mathcal{A}$ are the state and action spaces, respectively. The transition probability over the next state $s_{t+1} \in \mathcal{S}$ given the current state $s_t \in \mathcal{S}$ and action $a_t \in \mathcal{A}$ is denoted as $p(s_{t+1} \mid s_t, a_t)$, the initial state distribution as $p(s_1)$. A policy $\pi$ specifies a probability distribution over actions given a current state $s_t$. We now define the distribution over trajectory $\tau = (s_1, a_1, s_2, a_2, ..., s_T, a_T)$ for a sampling policy $\pi$ as $p_\pi(\tau) = p(s_1) \prod_t p(s_{t+1} \mid s_t, a_t)\pi(a_t \mid s_t)$. The reward function is given by $r(s_t, a_t) \in \mathbb{R}$. The standard objective in RL is to find a policy that maximizes expected return $\pi^* = \arg\max_\pi \mathbb{E}_{p_\pi(\tau)}[\sum_{t=1}^T r(s_t, a_t)]$. Solving this control problem becomes prohibitive in high dimensional and continuous state-action spaces. Formulating the RL problem as probabilistic inference in a graphical model instead allows the development of a variety of approximate inference algorithms for control in such settings.

### 2.1 POLICY OPTIMIZATION VIA PROBABILISTIC INFERENCE

The distribution $p_\pi(\tau)$ defines a generative process over trajectories, but it has no mechanism to distinguish between preferred trajectories with higher return. We incorporate the reward into the probabilistic model by introducing a set of binary auxiliary variables $\mathcal{O}_t \in \{0, 1\}$ that are *independently* distributed at each time as $p(\mathcal{O}_t = 1 \mid s_t, a_t) \propto \exp(r(s_t, a_t))$. The event $\mathcal{O}_t = 1$ can loosely be interpreted as having acted optimally at time $t$[1]. A natural objective is to find the policy that maximizes the log-probability of generating an "optimal" trajectory:

$$\max_\pi \log p_\pi(\mathcal{O}_{1:T}) = \max_\pi \log \mathbb{E}_{p_\pi(\tau)}\left[\exp\left(\sum_t r(s_t, a_t)\right)\right], \qquad (1)$$

---

[1]The optimality interpretation is a loose one stemming from the reward at time $t$, which increases the probability of $\mathcal{O}_t = 1$ exponentially. This interpretation has become standard in the RL-as-inference literature (Levine, 2018).

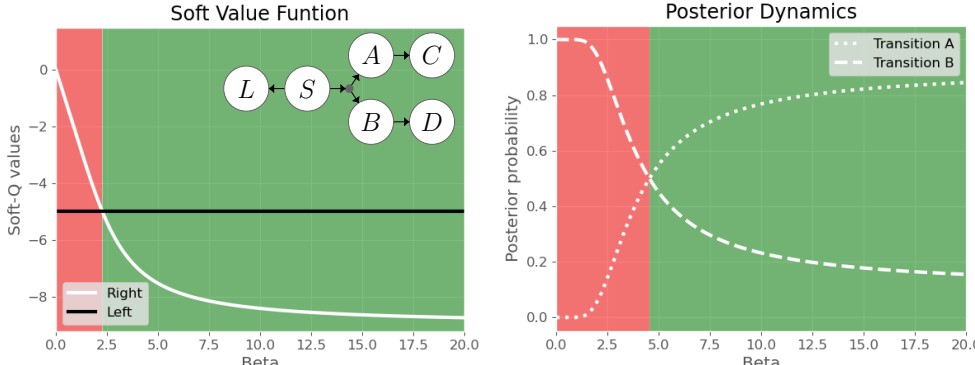

Figure 1: **Risky arm environment** *Left:* We compute the Soft-Q values as a function of $\beta$ for an MDP with 2 actions (left and right) and initial state S. Action 'right' produces no reward and a transition to either state A or B, with probability 0.9 and 0.1, respectively. State A and B have only one action that produces a deterministic transition to C and D with rewards -10 and 0, respectively. Action 'left' has a deterministic transition to L with reward -5. We illustrate the role of $\beta$ at modulating risk by observing that for small $\beta$ RL-as-Inference learns a risky policy (red region) while for large $\beta$ it recovers an optimal risk neutral policy (green region). *Right:* Similarly, $\beta$ modulates the resulting posterior transitions for A and B. Small $\beta$ produces risky dynamics (red region) while large $\beta$ recovers the true environment dynamics (green region).

where $\mathcal{O}_{1:T}$ is shorthand for $\mathcal{O}_t = 1$ for all times $t = 1, \ldots, T$. This log-marginal likelihood can be calculated via Bellman-style backup equations,

$$V_\pi(s_t) = \log \mathbb{E}_{\pi(a_t|s_t)}[\exp(Q_\pi(s_t, a_t))] \tag{2}$$

$$Q_\pi(s_t, a_t) = r(s_t, a_t) + \log \mathbb{E}_{p(s_{t+1}|s_t, a_t)}[\exp(V_\pi(s_{t+1}))]. \tag{3}$$

These are known as *soft* value functions, due to the presence of operators $\log \mathbb{E}[\exp(\cdot)]$ that act as soft approximations to $\max(\cdot)$. The log-marginal likelihood is then given by $\log p(\mathcal{O}_{1:T}) = \log \mathbb{E}_{p(s_1)}[\exp(V_\pi(s_1))]$.

## 2.2 OPTIMISTIC DYNAMICS AND POLICY RISK

The construction in Sec. 2.1 can exhibit risk-seeking behavior in the learned policy. In fact, the controller can be shown to optimize a policy under the assumption that dynamics are equal to the posterior distribution over transitions (Levine, 2018):

$$p(s_{t+1} \mid s_t, a_t, \mathcal{O}_{1:T}) \propto p(s_{t+1} \mid s_t, a_t) \exp(V_\pi(s_{t+1})). \tag{4}$$

This can be seen as an *exponential twisting* of the true dynamics by the soft-value functions (Chow et al., 2021). In stochastic environments the soft-value functions tend to be dominated by the maximum achievable reward while ignoring the probability of transitions. This reward-seeking behavior yields overly-optimistic dynamics and risk-seeking policies. Scaling rewards by a constant factor $\beta$ modulates this risk through the corresponding soft Bellman backups:

$$Q_\pi(s_t, a_t) = \frac{r(s_t, a_t)}{\beta} + \log \mathbb{E}_{p(s_{t+1}|s_t, a_t)}[\exp(V_\pi(s_{t+1}))]. \tag{5}$$

The state value function $V_\pi(s_t) = \log \mathbb{E}_{\pi(a_t|s_t)}[\exp(Q_\pi(s_t, a_t))]$ remains unchanged from Eq. (2) but is computed on the scaled action-value function. Note that applying a constant scale factor to rewards does not change the optimal policy in a classic RL setting, where one aims to maximize expected return. However, the factor $\beta$ nonlinearly scales terms in the log-marginal of Eq. (5) and therefore shifts the optimal policy. Fig. 1 illustrates the impact of $\beta$ on the optimal policy in a simple *risky arm* MDP. Optimizing the marginal likelihood in Eq. (1) is equivalent to $\beta = 1$ and results in the agent choosing an optimistic action (right arm). However, for $\beta > 2.5$ we recover a policy that maximizes expected return as in the classic RL setting. We also observe that $\beta$ modulates the posterior dynamics (Eq. (4)), as for $\beta > 5$ the posterior dynamics start to converge to the real dynamics.

# 3 RISK SENSITIVE VARIATIONAL MODEL-BASED POLICY OPTIMIZATION

The previous section showed that RL-as-inference can lead to risk-seeking policies. This property also arises in model-based variational methods, where posterior dynamics must be approximated. In this section, we discuss adaptive risk in this approximate setting and present the $\beta$-VMBPO algorithm for RL under adaptive policy risk. Pseudocode for the algorithm can be found in the Appendix.

## 3.1 RISK-ADAPTIVE EVIDENCE LOWER BOUND (ELBO)

In Sec. 2.2 we showed that scaling the rewards by a factor $\beta$ modulates the policy risk through the assumed posterior dynamics. The corresponding soft-value functions yield a log-marginal likelihood for this re-scaled model:

$$\beta \log p_\pi(\mathcal{O}_{1:T}; \beta) = \beta \log \mathbb{E}_{p_\pi(\tau)} \left[ \exp \left( \frac{\sum_t r(s_t, a_t)}{\beta} \right) \right]. \tag{6}$$

We have scaled both sides by a factor $\beta$ to show that the r.h.s. of Eq. (6) is the well-known *risk sensitive exponentiated utility*[2] from Safe RL (García & Fernández, 2015). The risk-sensitive log-marginal lacks a closed-form so we consider a variational approach with the risk-sensitive ELBO:

$$\beta \log p_\pi(\mathcal{O}_{1:T}; \beta) \geq \max_q \mathbb{E}_{q(\tau)} \left[ \sum_t r_t(s_t, a_t) \right] - \beta \mathrm{KL}(q(\tau) \,\|\, p_\pi(\tau)). \tag{7}$$

where $q(\tau) = p(s_1) \prod_t q(s_{t+1} \mid s_t, a_t) q(a_t \mid s_t)$. From Eq. (7) it is clear how risk-seeking arises as the expectation is taken w.r.t. the variational distribution, which approaches the posterior dynamics $q(s_{t+1} \mid s_t, a_t) \approx p(s_{t+1} \mid s_t, a_t, \mathcal{O}_{1:T})$. Although the penalty $\mathrm{KL}(q(\tau) \,\|\, p_\pi(\tau))$ discourages large deviations from the prior, the controller may learn optimistic dynamics that do not reflect those of the true environment. Observe that Eq. (7) reflects the Lagrangian of the following constrained optimization,

$$\max_q \mathbb{E}_{q(\tau)} \left[ \sum_t r(s_t, a_t) \right] \text{ s.t. } \mathrm{KL}(q(\tau) \,\|\, p_\pi(\tau)) \leq \epsilon. \tag{8}$$

The use of $\epsilon$ suggests that the allowable divergence should be small, to ensure that the learned policy is applicable in the real environment. Model-free methods such as *soft actor critic* (SAC) remove this penalty and impose the hard constraint that $q(s_{t+1} \mid s_t, a_t) = p(s_{t+1} \mid s_t, a_t)$, namely variational dynamics equal the true environment dynamics (Haarnoja et al., 2017; 2018; Levine, 2018). With this constraint model terms cancel in the resulting objective, which is known as MaxEnt RL.

**Dual Optimization** Our approach allows tighter bounds than MaxEnt RL by learning the allowable deviation through the $\beta$ risk parameter. For this we recognize $\beta$ as a Lagrange multiplier and perform dual-descent via the loss function:

$$J(\beta) = \beta \epsilon - \beta \mathrm{KL}(q(s_{t+1} \mid s_t, a_t) \,\|\, p(s_{t+1} \mid s_t, a_t)). \tag{9}$$

Observe that the constraint in the primal problem of Eq. (8) suggests optimizing the dual parameter $\beta$ w.r.t. the entire trajectory $\mathrm{KL}(q(\tau) \,\|\, p(\tau))$. This can lead to high variance for long trajectories. We instead impose the constraint only on the variational dynamics, which yields more stable learning. We emphasize that the role of risk parameter $\beta$ is not to perform Safe RL, but rather is a dual variable that is adaptively learned with the policy and value functions online.

## 3.2 DYNAMICS OPTIMIZATION

We model the prior $p_\theta$ and variational $q_\phi$ dynamics using an ensemble of probabilistic neural networks such that outputs of each member parameterize a Gaussian. This model has been widely used in model-based RL for its ability of handling uncertainty that can reduce policy exploitation (Janner et al., 2019). We also distinguish between samples coming from interactions with the real environment $\mathcal{D}_{\mathrm{env}}$ and samples coming from our variational model $\mathcal{D}_{\mathrm{model}}$. We learn the parameters for the prior dynamics $p_\theta$ using only samples from the environment by minimizing the cross-entropy loss,

$$J(\theta) = -\mathbb{E}_{(s_t, a_t, s_{t+1}, r_t) \sim \mathcal{D}_{\mathrm{env}}} \left[ \log p_\theta(s_{t+1}, r_t \mid s_t, a_t) \right], \tag{10}$$

---

[2] A related discussion is given in the Appendix of Eysenbach et al. (2022).

where $r_t = r(s_t, a_t)$. To learn the parameters for the variational dynamics $q_\phi$, we observe that the lower bound is tight when the variational dynamics are equal to the posterior dynamics Eq. (4). Hence, we minimize the forward KL divergence $\mathrm{KL}(p(s_{t+1} \mid s_t, a_t, \mathcal{O}_{1:T}) \,\|\, q_\phi(s_{t+1} \mid s_t, a_t))$, which simplifies to:

$$J(\phi) = -\mathbb{E}_{(s_t, a_t, s_{t+1}, r_t) \sim \mathcal{D}_{\mathrm{env}}} \left[ \exp\left( \frac{1}{\beta}(r_t + V'_\psi(s_{t+1}) - Q_\psi(s_t, a_t)) \right) \log q_\phi(s_{t+1} \mid s_t, a_t) \right],$$
(11)

In stochastic environments this objective can be risk seeking as it inflates the densities where a transition to $s_{t+1}$ is unlikely but the target $r_t + V'_\psi(s_{t+1})$ is larger than $Q_\psi(s_t, a_t)$. Conversely, in deterministic environments it exponentially prioritizes learning transitions where the TD error is positive, which leads to risk-seeking. Our risk parameter $\beta$ directly modulates this behaviour prioritizing transitions with high potential without complete blindness to unlikely events. Both Eq. (10) and (11) are optimized by stochastic gradient descent using samples from the experience buffer $\mathcal{D}_{\mathrm{env}}$.

### 3.3 ACTOR-CRITIC OPTIMIZATION

We approximate the soft-critics $V_\psi$ and $Q_\psi$ in Eq. (11) using function approximators for an MDP with variational dynamics $q_\phi$, variational policy $q_\omega$, and augmented rewards $r(s_t, a_t) - \beta \log \frac{q_\phi(s_{t+1}|s_t, a_t)}{p_\theta(s_{t+1}|s_t, a_t)}$ where we estimate the log term using our two learned dynamics models, $p_\theta$ and $q_\phi$. We learn the parameters for the critic $Q_\psi$ by minimizing the squared TD error:

$$J(\psi) = \mathbb{E}_{(s_t, a_t, s_{t+1}, r_t) \sim \mathcal{D}_{\mathrm{model}}} \left[ \left( Q_\psi(s_t, a_t) - r_t - V'_\psi(s_{t+1}) + \beta \log \frac{q_\phi(s_{t+1} \mid s_t, a_t)}{p_\theta(s_{t+1} \mid s_t, a_t)} \right)^2 \right],$$
(12)

which we optimize by stochastic gradient descent using samples from our variational replay buffer — we produce rollouts from our variational dynamics and policy and store them in a replay buffer $\mathcal{D}_{\mathrm{model}}$ — and $V_\psi(s_{t+1})$ is implicitly represented by $\mathbb{E}_{a_{t+1} \sim q_\omega(a_{t+1}|s_{t+1})}[Q'_\psi(s_{t+1}, a_{t+1})]$ which we approximate by sampling from the posterior policy $q_\omega$.

We model the prior and variational policy using Gaussian distributions with their mean and variance represented by neural networks, $\pi_\kappa$ and $q_\omega$. To learn the parameters for the variational policy $q_\omega$, we observe that the ELBO in Eq. (7) is tight when the variational policy is equal to the posterior policy, $p(a_t|s_t, O_{1:T}) \propto \pi(a_t|s_t) \exp(Q(s_t, a_t))$. Hence, we minimize the backward KL divergence $\mathrm{KL}(q_\omega(a_t \mid s_t) \,\|\, p(a_t \mid s_t, \mathcal{O}_{1:T}))$:

$$J(\omega) = \mathbb{E}_{s_t \sim \mathcal{D}_{\mathrm{model}}} \left[ \mathbb{E}_{a_t \sim q_\omega(\cdot|s_t)} \left[ \log q_\omega(a_t|s_t) - Q'_\psi(s_t, a_t) - \log \pi_\kappa(a_t|s_t) \right] \right].$$
(13)

This expectation is taken w.r.t. the variational policy. We use the reparameterization trick to obtain a lower variance estimator. The prior policy $\pi_\kappa$ acts as a trust region constraint on the optimization steps taken by the variational policy. Optimization of the ELBO Eq. (7) w.r.t. the prior policy $\pi_\kappa$ only involves the term $\mathrm{KL}(q_\omega \,\|\, \pi_\kappa)$, so the loss function is:

$$J(\kappa) = -\mathbb{E}_{(s_t, a_t) \sim \mathcal{D}_{\mathrm{model}}} \left[ \log \pi_\kappa(a_t|s_t) \right].$$
(14)

In our implementation, we use the same network architectures to represent, both the prior and posterior policy. Hence, the KL can be minimized by setting the parameters of the network for $\pi_\kappa$ equal to those of $q_\omega$.

## 4 THEORETICAL ANALYSIS

Model-based algorithms for RL-as-inference lack the fundamental theoretical analysis of classical methods. We provide a thorough theoretical analysis of our risk-sensitive Bellman operator beginning with a policy evaluation theorem that shows the Bellman operator is a contraction (Theorem 4.1). Our policy improvement Theorem 4.2 shows the greedy policy monotonically improves value function estimates. We then establish a policy iteration theorem showing that repeated application of policy evaluation and improvement converges to an optimal deterministic policy (Theorem 4.3). Finally, connecting the theory to our algorithm, we show that the $\beta$-VMBPO objective has an optimum at a pair of policies under the prior and posterior that are deterministic and equivalent

(Theorem 4.4). All proofs can be found in the Appendix. Our analysis begins with the risk-sensitive Bellman operator $\mathcal{T}$:

$$\mathcal{T}Q(s_t, a_t) = \frac{r(s_t, a_t)}{\beta} + \gamma \log \mathbb{E}_{s_{t+1}, a_{t+1} \sim p_\pi(\cdot|s_t, a_t)} \left[ \exp(Q(s_{t+1}, a_{t+1})) \right]. \tag{15}$$

Starting with some random function $Q^0$ we iteratively apply this operator, $\mathcal{T}Q^k = Q^{k+1}$. Using the following theorem, we conclude that the operator is a contraction, and repeated application converges to the true value function.

**Theorem 4.1.** *(Risk Sensitive Policy Evaluation) Let $\mathcal{T}$ be the risk sensitive operator w.r.t. some policy $\pi$, $\mathcal{T}Q(s_t, a_t) = \frac{r(s_t, a_t)}{\beta} + \gamma \log \mathbb{E}_{s_{t+1}, a_{t+1} \sim p_\pi(\cdot|s_t, a_t)}[\exp(Q(s_{t+1}, a_{t+1}))]$. Then $\mathcal{T}$ is a contraction and $\lim_{k \to \infty} Q^k = Q_\pi$.*

Once the iterative process has converged we have obtained the action-value functions $Q_\pi$. Now we present the policy improvement step which monotonically produces a better policy $\pi'$. We consider the greedy policy w.r.t. the $Q_\pi$. This new policy is guaranteed to be at least as good as $\pi$ by,

**Theorem 4.2.** *(Risk Sensitive Policy Improvement) Let $\pi$ be a policy and $\pi'$ the greedy policy w.r.t. the action-value function $Q_\pi$, $\pi'(s_t) = \arg\max_{a_t} Q_\pi(s_t, a_t)$. Then, $V_{\pi'}(s_t) \geq V_\pi(s_t)$ for all states $s_t \in \mathcal{S}$.*

Combining our two results leads to a policy iteration algorithm composed of alternating between $Q_\pi$ estimation and policy improvement by it taking the greedy policy w.r.t. $Q_\pi$. This process stops when the greedy policy does not produce an improvement. Moreover, the process converges to a *deterministic* policy.

**Theorem 4.3.** *(Risk Sensitive Policy Iteration) Given any initial policy $\pi$ repeated application of risk sensitive policy evaluation (Thm. 4.1) and policy improvement (Thm. 4.2) converges to a deterministic optimal policy $\pi^*$.*

The previous theorem guarantees that there exist a deterministic optimal policy $\pi^*$ as we can always obtain one by producing a policy from any optimal $Q^*$ function. But we also know that w.r.t. any policy $\pi$ the optimal variational policy has the form $q(a_t|s_t) \propto \pi(a_t|s_t) \exp(Q(s_t, a_t))$. Therefore, $q(a_t|s_t)$ must be equal to the optimal policy $\pi^*(a_t|s_t)$.

**Theorem 4.4.** *There exist an optimal pair of policies $\pi^*$ and $q^*$ which are equal and deterministic for the $\beta$-VMBPO objective:*

$$\arg\max_{\pi, q} \mathbb{E}_{q(\tau)} \left[ \frac{\sum_t r(s_t, a_t)}{\beta} \right] - \mathrm{KL}(q(\tau) \,\|\, p_\pi(\tau))$$

Related results for a variant of the soft-Q-learning policy evaluation are presented in Fox et al. (2016) and Haarnoja et al. (2017). While related those results apply in the MaxEnt RL framework and differ from our more general setting. Given that soft-updates is commonly used to refer to soft-Q learning policy iteration we distinguish our results with the "Risk Sensitive" nomenclature.

## 5 RELATED WORK

The duality between optimal control and posterior inference dates back to Kalman (1960). Todorov (2006) showed that this duality holds for the class of Linearly solvable MDPs. Levine & Koltun (2013); Levine (2018) showed RL can be formulated as inference with the formulation that we use in this work, but inference is intractable in general (Todorov, 2008). Variational approximate RL-as-inference by maximizing a lower bound on the marginal likelihood. Maximum likelihood policies for the PGM formulation correspond to policies that maximize the exponential utility (Eysenbach et al., 2022; Noorani & Baras, 2022) — a utility with a long history in the risk-sensitive control literature (García & Fernández, 2015; Von Neumann & Morgenstern, 1947; Jacobson, 1973; Mihatsch & Neuneier, 2002). This approach simultaneously maximizes expected return and variance, resulting in risk-seeking policies (Levine, 2018; Depeweg et al., 2018).

Maximum Entropy RL avoids risk-seeking behavior by removing the controller's ability to modify the posterior dynamics, resulting in model-free learning and high-entropy policies. This penalization of determinism has shown effective in high-dimensional tasks (Ziebart et al., 2008; Haarnoja

et al., 2017; 2018). However, stochastic MaxEnt RL policies can lead to undesirable behavior as established by Fellows et al. (2019). Another example is that of KL-regularized RL where a penalty is included for policies that diverged from an old policy (Schulman et al., 2015; 2017; Noorani & Baras, 2021). These problems can also be optimized by EM-like algorithms, which alternate optimization of the posterior and prior policies (Abdolmaleki et al., 2018b;a).

Our work generalizes VMBPO (Chow et al., 2021) allowing for more flexible variational approximations. All previously-mentioned methods require tuning of a hyperparameter similar to our $\beta$ parameter—typically learned through costly repeated training runs. Our $\beta$-VMBPO approach is most similar to the dual optimization in SAC (Haarnoja et al., 2018). However, the $\beta$ parameter in our present work directly impacts policy risk through the risk-sensitive exponentiated utility function. The same is not true for SAC, where the hyperparameter serves as a weighting coefficient controlling policy entropy. Our work is the first to explore the role of $\beta$ as a risk parameter in model-based RL-as-inference, propose a mechanism for tuning this parameter simultaneously with agent learning, and to provide an open-source implementation of VMBPO and our extension $\beta$-VMBPO.

## 6 EXPERIMENTS

The goals of our experiments are threefold: first, to study the impact of risk parameter $\beta$ on modulating optimism in the posterior dynamics; second, verify robustness of our dual optimization; third, evaluate performance of $\beta$-VMBPO on benchmark reinforcement learning tasks compared to other model-based and RL-as-inference algorithms. We were unable to obtain source code associated with the original VMBPO paper Chow et al. (2021). Thus our experiments are based on a reimplementation of VMBPO with every attempt to respect the original algorithm. Our code will be made available in the supplement and will be open-source with publication.

### 6.1 RISK IN TABULAR ENVIRONMENTS

We begin with an examination of the gridworld environment presented in Eysenbach et al. (2022). We modify the original environment to incorporate risk by including a cliff region. Falling into the cliff incurs a large negative reward and transition to the initial state. The agent can choose from four actions (up, left, down and right) which can result in a transition to the chosen direction or moving randomly to one of the four directions with equal probability (See Fig. 2a). We first consider the posterior dynamics and policies learned for two fixed $\beta$ values, Fig. 2b and 2c. The dynamics learned under small $\beta$ demonstrate an alarming behaviour — transitions almost deterministically lead toward the goal. Then the posterior policy exploits this belief and chooses to move as close as possible to the cliff ignoring any chance of falling down. In contrast, the posterior dynamics under large $\beta$ obey the prior dynamics which allows the agent to learn a safer policy that reaches the goal.

Now, we test the performance and robustness of our algorithm $\beta$-VMBPO. We consider two tabular algorithms: VMBPO (Chow et al., 2021) which is equivalent to our algorithm for fixed $\beta = 1$, and Q-learning (Watkins & Dayan, 1992). We defer comparison to Mismatch No More (MnM) (Eysenbach et al., 2022) to the Appendix as negative rewards are a limitation of that method. Our algorithm outperforms both methods by converging faster to the goal (Fig. 3a). To test robustness, we initialize $\beta$ at different values. For all our experiments, our algorithm learns a near-optimal policy (Fig. 3b) and our risk parameter converges to the same value (Fig. 3c).

### 6.2 OPTIMISM IN DETERMINISTIC CONTINUOUS ENVIRONMENTS

The use of function approximators can lead to unrealistically optimistic dynamics, even in deterministic environments. Note that in non-stochastic environments the KL penalty (Eq. 7) forces posterior dynamics to equal the true prior dynamics. However, the use of flexible function approximators can fail to learn realistic dynamics without adapting the $\beta$ parameter during learning. To demonstrate this behavior we consider Continuous Mountain Car; a low-dimensional and continuous environment where a car is positioned randomly over a valley. The car must reach the top of the hill, but is under powered and so must build momentum. This experiment presents an opportunity to introduce function approximators into the optimization of $\beta$-VMBPO, but keeps the problem simple enough to study the effects of $\beta$ over the learning algorithm.

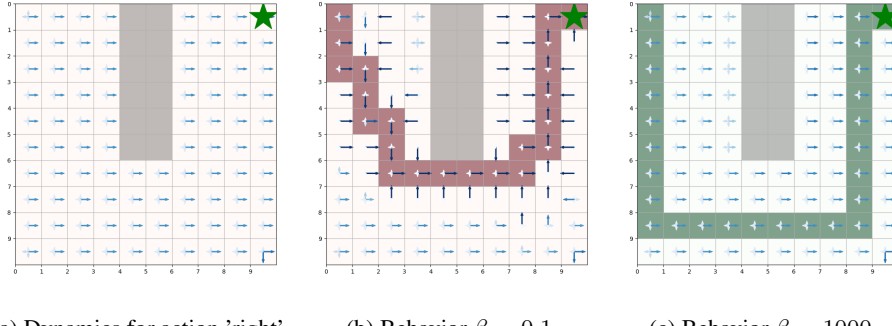

(a) Dynamics for action 'right'.    (b) Behavior $\beta = 0.1$.    (c) Behavior $\beta = 1000$.

Figure 2: *Left*: In this figure, we show the real dynamics in the grid for the action 'right' at each state, where we represent a transition probability between two states as a vectors with its magnitude proportional to the probability. *Middle*: Small $\beta$ leads to variational dynamics that are close to deterministic and ignore the prior direction of the dynamics. Policies learn under these dynamics ignore the risk of falling off the cliff (red path). *Right*: Large $\beta$ leads to variational dynamics that imitate the real environment. Hence, policies learn under these dynamics can be optimal (green path).

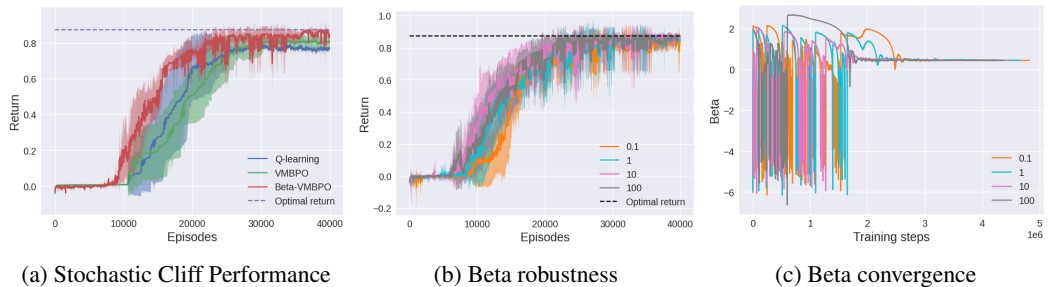

(a) Stochastic Cliff Performance    (b) Beta robustness    (c) Beta convergence

Figure 3: *Left*: We compare the expected return for 5 independent runs for different algorithms. $\beta$-VMBPO outperforms both Q-learning and VMBPO both by learning the task quicker and finding a better return even after 40000 episodes. *Middle*: We demonstrate that our algorithm is robust to $\beta$ initialization. Leading to high return over the course of learning. *Right*: We show that $\beta$ optimization eventually converges to the same optimal value for the different initialization.

As shown in Fig. 4, optimizing VMBPO can result in a optimistic environment where the agent ignores the constraint of momentum leading to bad policies. We claim that this risk-seeking behaviour comes as the result of optimizing Eq. (12). Transitions with high positive TD error are exponentiated, forcing the approximator to prioritized them in the loss function. $\beta$-VMBPO resolves these issues by finding some appropiate regularization and re-scaling these TD-error. Hence, the learned model produces similar trajectories to those from the real environment (Fig. 4).

## 6.3 HIGH-DIMENSIONAL BENCHMARKS

We compare our method to two RL-as-inference algorithms, SAC (Haarnoja et al., 2018), a model-free RL algorithm that optimizes a trade-off between return and policy entropy and model-based VMBPO (Chow et al., 2021), which optimizes a risk-seeking objective ($\beta = 1$). This creates a nice comparison to our method's risk modulation via $\beta$ optimization. Finally, we consider a well-established baseline for model-based RL algorithms, MBPO (Janner et al., 2019), an algorithm that also learns a maximum likelihood dynamics model. For the latter method, we have modified the optimization of its dynamics model to a fixed number of iterations (For more implementation details see the Appendix).

We evaluate these baselines on three high-dimensional continuous tasks from OpenAI gym benchmark suite (Hopper, Walker2D, HalfCheetah). For consistency we use the same network architectures and update schedule across all algorithms including the actor-critic and dynamics model. Similarly, we have the same schedule to generate model rollouts. We perform five runs of each algorithm with different random seeds and report average and STDEV every 1k environment steps.

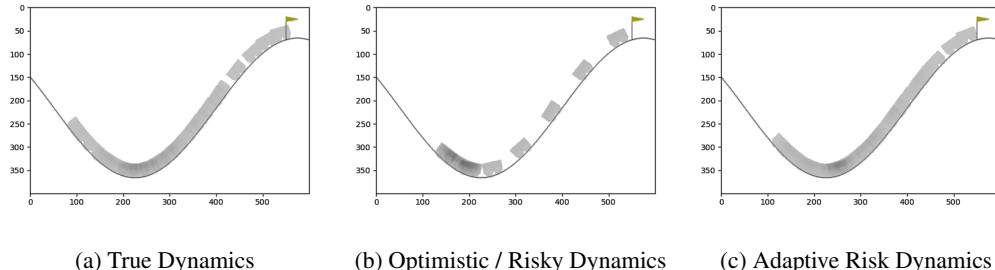

(a) True Dynamics  (b) Optimistic / Risky Dynamics  (c) Adaptive Risk Dynamics

Figure 4: **Mountain Car Variational Dynamics.** We compute an average over trajectories for different learned dynamics and policies. *Left*: This figure illustrates the known strategy for this game by using the real dynamics and a learned policy. The car generates momentum by moving back and forward until it can reach the goal. *Middle*: This figure uses an optimistic dynamics learned by VMBPO ($\beta = 1$). The agent learns a policy in which it accelerates directly into the goal. Unfortunately, applying this policy in the real environment will result in the car getting stuck in the mountain. *Right*: Finally, we have the dynamics learned by our $\beta$-VMBPO. These dynamics respect the constraints impose by prior dynamics, and thus policies learned under these dynamics can be transferred into the original environment.

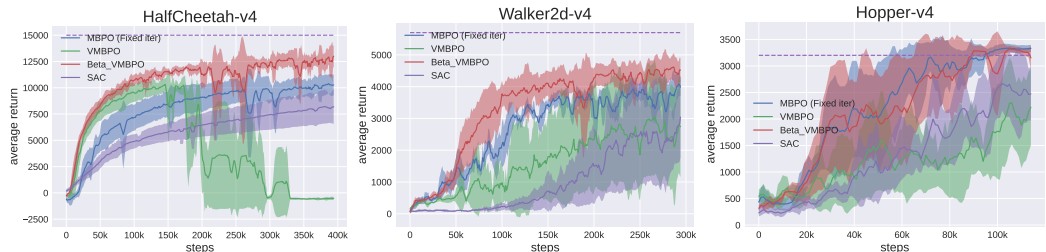

Figure 5: Training curves of $\beta$-VMBPO and other baselines for OpenAI Gym benchmarks. The solid curves correspond to the mean and shaded regions to the standard deviation over 5 random trials. $\beta$-VMBPO outperforms the other methods in HalfCheetah and Walker2d while performing consistently in the simplest task, Hopper. We also provide asymptotic SAC performance (dashed line) which requires many more environment interactions (in the millions) compared to our more sample-efficient methods.

The results show that $\beta$-VMBPO outperformed VMBPO across all tasks (Fig. 5). One can attribute VMBPO's subpar performance to the degradation of its variational dynamics — the exponential-TD error in its dynamics objective can result in instabilities towards the end of learning. In contrast, our method modulates this exponential term resulting in stable learning. $\beta$-VMBPO also outperformed MBPO (Fixed iterations) in both HalfCheetah and Walker2D environments demonstrating that $\beta$ modulation can improve learning by prioritizing certain transitions using its variational model. Performance is similar between MBPO (Fixed iterations) and $\beta$-VMBPO for Hopper–we hypothesize that this task is near saturation as evidenced by the asymptotic performance of SAC.

## 7 CONCLUSION

We provided a comprehensive exploration of risk-ssensitivity in model-based RL-as-inference along with a practical algorithm, $\beta$-VMBPO, that adaptively modulates risk during learning. Our experimental results support our claims that $\beta$-VMBPO is effective in learning for a range of tasks and yields superior sample efficiency to model-free baselines. Most importantly, we show uniform improvement over baseline VMBPO when accurately matching the published configuration of that work. Our theoretical analysis establishes fundamental properties of RL-as-inference learning in the risk-sensitive regime. We emphasize that our general theoretical results specialize to the standard RL-as-inference regime when setting $\beta = 1$. A limitation of our methodology is that one is required to tune the constraint parameter $\epsilon$. However, unlike traditional regularization coefficients, $\epsilon$ has a straightforward interpretation of allowable posterior deviation (in nats) and can be more easily tuned. Furthermore, we find performance less sensitive to this parameter than the alternative of tuning $\beta$ as a regularization coefficient. All code for this work will be publicly accessible upon publication.

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

APPENDIX

## A  RISK SENSITIVE POLICY ITERATION PROOFS

**Theorem 4.1.** *(Risk Sensitive Policy Evaluation) Let $\mathcal{T}$ be the risk sensitive operator w.r.t. some policy $\pi$, $\mathcal{T}Q(s_t, a_t) = \frac{r(s_t, a_t)}{\beta} + \gamma \log \mathbb{E}_{s_{t+1}, a_{t+1} \sim p_\pi(\cdot|s_t, a_t)}[\exp(Q(s_{t+1}, a_{t+1}))]$. Then $\mathcal{T}$ is a contraction and $\lim_{k \to \infty} Q^k = Q_\pi$.*

*Proof.* Let $Q_1$ and $Q_2$ be two action value functions. We define $\epsilon = ||Q_1 - Q_2||_\infty = \max_{(s_t, a_t) \in \mathcal{S} \times \mathcal{A}} |Q_1(s_t, a_t) - Q_2(s_t, a_t)|$. Then for all pairs $(s_t, a_t)$ we have that, $Q_1(s_t, a_t) \leq Q_2(s_t, a_t) + \epsilon$. Then,

$$\log \mathbb{E}_{s_{t+1}, a_{t+1} \sim p_\pi} [\exp(Q_1(s_{t+1}, a_{t+1}))] \leq \log \mathbb{E}_{s_{t+1}, a_{t+1} \sim p_\pi} [\exp(Q_2(s_{t+1}, a_{t+1}) + \epsilon)]$$
$$= \log \mathbb{E}_{s_{t+1}, a_{t+1} \sim p_\pi} [\exp(Q_2(s_{t+1}, a_{t+1}))] + \epsilon,$$

where we use the monotonicity of the $\exp$, expectation and $\log$ operations. Analogously, we have that $\log \mathbb{E}_\pi [\exp(Q_2(s_t, a_t))] \leq \log \mathbb{E}_\pi [\exp(Q_1(s_t, a_t))] + \epsilon$. Therefore, $||\mathcal{T}Q_1 - \mathcal{T}Q_2||_\infty \leq \gamma \epsilon = \gamma ||Q_1 - Q_2||_\infty$. This proof generalizes a result by Haarnoja et al. (2017); Fox et al. (2016) for the soft-Q learning framework. □

**Lemma 4.1.** *(Optimal Risk Sensitive Operator) Let $\mathcal{T}^*$ be the optimal risk sensitive operator, $\mathcal{T}^*Q(s_t, a_t) = \frac{r(s_t, a_t)}{\beta} + \gamma \log \mathbb{E}_{s_{t+1} \sim p(\cdot|s_t, a_t)}[\max_{a_{t+1}} \exp(Q(s_{t+1}, a_{t+1}))]$. Then $\mathcal{T}^*$ is a contraction.*

*Proof.* The proof follows the same argument as Theorem 4.1, but the expectation over $s_{t+1}$ and $a_{t+1}$ is replaced by an expectation over $s_{t+1}$ and maximization over $a_{t+1}$. These operations are also monotonic so the result follows. □

**Theorem 4.2.** *(Risk Sensitive Policy Improvement) Let $\pi$ be a policy and $\pi'$ the greedy policy w.r.t. the action-value function $Q_\pi$, $\pi'(s_t) = \arg\max_{a_t} Q_\pi(s_t, a_t)$. Then, $V_{\pi'}(s_t) \geq V_\pi(s_t)$ for all states $s_t \in \mathcal{S}$.*

*Proof.* We start by assuming a finite-horizon problem. We prove the improvement by induction. We consider the base case, $V_\pi(s_T) \leq V_{\pi'}(s_T)$, where $s_T$ is the last state on an episode. By definition, $Q_\pi(s_T, a_T) = r(s_T, a_T) = Q_{\pi'}(s_T, a_T)$ for all actions $a_T$. Now we prove the base,

$$V_\pi(s_T) = \log \mathbb{E}_{a_T \sim \pi}[\exp Q_\pi(s_T, a_T)]$$
$$\leq \max_{a_T} Q_\pi(s_T, a_T)$$
$$= \log \mathbb{E}_{a_T \sim \pi'}[\exp Q_\pi(s_T, a_T)]$$
$$= \log \mathbb{E}_{a_T \sim \pi'}[\exp Q_{\pi'}(s_T, a_T)]$$
$$= V_{\pi'}(s_T),$$

where we use the fact that max operator bounds the softmax operator, and the equality, $Q_\pi(s_T, a_T) = Q_{\pi'}(s_T, a_T)$. We now demonstrate the induction step — $V_\pi(s_{t+1}) \leq V_{\pi'}(s_{t+1})$ implies $V_\pi(s_t) \leq V_{\pi'}(s_t)$. We have that

$$Q_\pi(s_t, a_t) = r(s_t, a_t) + \gamma \log \mathbb{E}_{s_{t+1} \sim p(\cdot|s_t, s_t)}[\exp V_\pi(s_{t+1})]$$
$$\leq r(s_t, a_t) + \gamma \log \mathbb{E}_{s_{t+1} \sim p(\cdot|s_t, s_t)}[\exp V_{\pi'}(s_{t+1})]$$
$$= Q_{\pi'}(s_t, a_t).$$

where we use the induction hypothesis. Thus, $Q_\pi(s_t, a_t) \leq Q_{\pi'}(s_t, a_t)$ for every actions $a_t$. Now we prove the induction step,

$$V_\pi(s_t) = \log \mathbb{E}_{a_t \sim \pi}[\exp Q_\pi(s_t, a_t)]$$
$$\leq \max_a Q_\pi(s_t, a)$$
$$= \log \mathbb{E}_{a_t \sim \pi'}[\exp Q_\pi(s_t, a_t)]$$
$$\leq \log \mathbb{E}_{a_t \sim \pi'}[\exp Q_{\pi'}(s_t, a_t)]$$
$$= V_{\pi'}(s_t).$$

where again we use that max bounds the softmax, and the inequality for $Q_\pi$ and $Q_{\pi'}$ shown in the previous statement. This completes our induction, showing that $V_{\pi'}(s_t) \geq V_\pi(s_t)$ for every state $s_t \in S$. □

**Theorem 4.3.** *(Risk Sensitive Policy Iteration) Repeated application of risk sensitive policy evaluation and policy improvement to any initial policy $\pi$ converges to a deterministic optimal policy $\pi^*$.*

*Proof.* Theorem 4.1 demonstrates that repeated application of the backup recovers the action-values $Q_\pi$. Theorem 4.2 demonstrates that the policy improves or remains fixed. Finally, Lemma 4.1 demonstrates that improvement reaches a fixed point at an optimal policy and the result holds. Note that Theorem 4.2 assumes a finite-horizon, thus the assumption carries to the present result. Extension to the infinite-horizon setting is being explored. □

**Theorem 4.4.** *There exist an optimal pair of policies $\pi^*$ and $q_c^*$ which are equal and deterministic for the $\beta$-VMBPO objective:*

$$\arg\max_{\pi,q} \mathbb{E}_{q(\tau)}\left[\frac{\sum_t r(s_t, a_t)}{\beta}\right] - \mathrm{KL}(q(\tau) \| p_\pi(\tau))$$

*Proof.* Theorem 4.3 demonstrates that there exist an optimal policy $\pi^*$. By applying policy improvement to an optimal policy we can always obtain a deterministic optimal policy. On the other hand, we know that the optimal variational policy is equal to the posterior policy, $p(a_t|s_t, \mathcal{O}_{1:T}) \propto \pi(a_t|s_t)\exp(Q_\pi(s_t, a_t))$ Chow et al. (2021). Therefore, a deterministic optimal policy $\pi^*$ has a determistic optimal variational policy $q^*(a_t|s_t)$. □

# B ADDITIONAL EXPERIMENTS

## B.1 TABULAR EXPERIMENTS

In Fig. 3a, we compare the performance of $\beta$-VMBPO to other algorithms in a tabular setting. We compare to VMBPO and Q-learning in the gridworld environment presented in Eysenbach et al. (2022). We also include Mismatched No More (MnM) Eysenbach et al. (2022), an algorithm that optimizes the same objective as VMBPO with log-transformed rewards. Particularly, we consider two different initializations of the Q-values (Fig. 6).

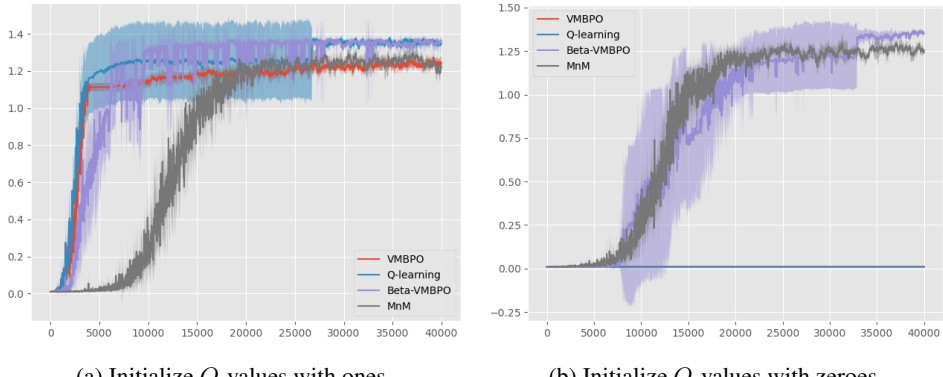

(a) Initialize $Q$-values with ones.  (b) Initialize $Q$-values with zeroes.

Figure 6: *Left*: This an optimistic initialization that results in high exploration — the reward is 0.001 for almost every state and action. Q-learning and $\beta$-VMBPO perform the best. We believe this is the result of VMBPO and MnM optimizing a sub-optimal objective. *Right*: This initialization is optimistic for MnM — the log-operation transforms the reward to negative — but pessimistic for the other methods. Q-learning and VMBPO struggle at learning any good policy, but $\beta$ modulation combats these effects by scaling its rewards. This permits $\beta$-VMBPO to learn an optimal policy that outperforms MnM's return.

## B.2    OPEN AI GYM BENCHMARK

In Fig. 7, we compare the performance of $\beta$-VMBPO to other algorithms in the environment Ant-v4 from OpenAI Gym benchmarks. Again we observe that $\beta$-VMBPO out-performs other baselines in this environment.

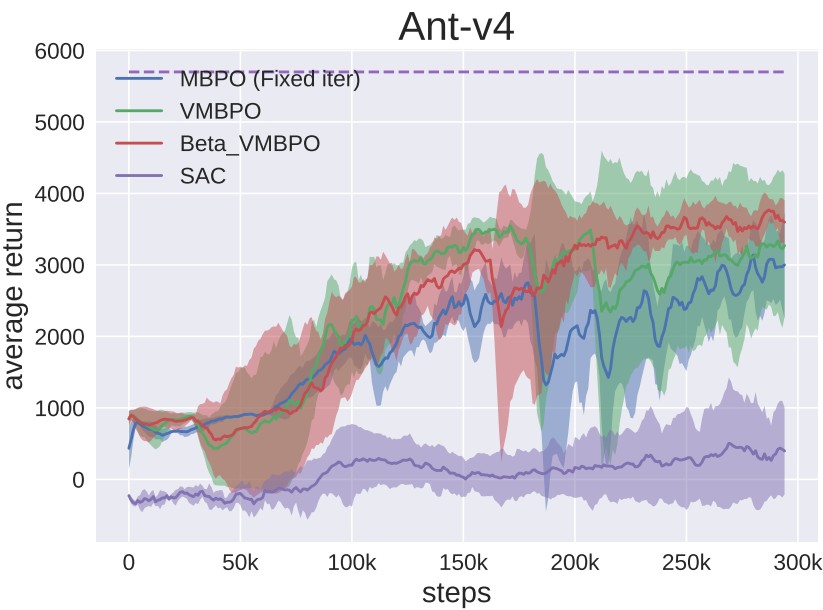

Figure 7: Training curves of $\beta$-VMBPO and other baselines for Ant-v4 from OpenAI Gym benchmarks. The solid curves correspond to the mean and shaded regions to the standard deviation over 5 random trials.

## B.3    ABLATION STUDY

In Fig. 8, we study the robustness of $\beta$-VMBPO w.r.t. the initialization and learning rate of $\beta$ in a high dimensional task (Hopper-v4). The solid curves correspond to the mean and shaded regions to the standard deviation over 5 random trials.

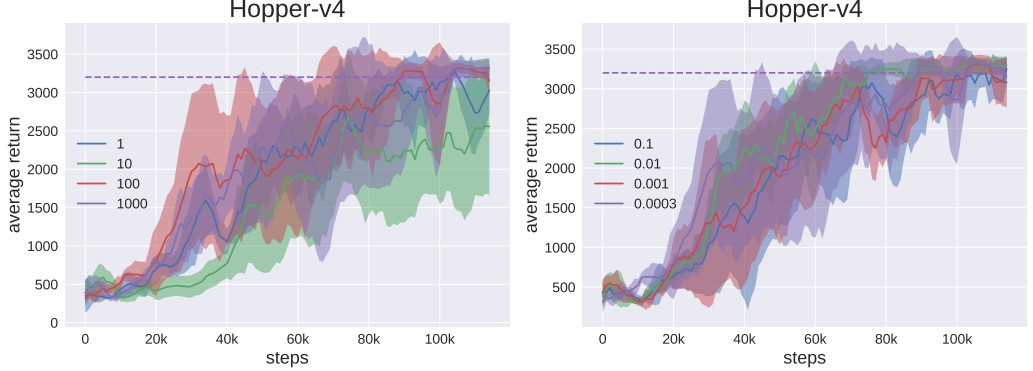

Figure 8: *Left*: We demonstrate that our algorithm is robust to $\beta$ initialization (1, 10, 100, 1000). *Right*: We demonstrate that our algorithm is robust to different learning rates for $\beta$ optimization (0.1, 0.01, 0.001, 0.0003).

## C   IMPLEMENTATION DETAILS

### C.1   NETWORK ARCHITECTURES

We implement VMBPO on top of Pytorch reimplementations for MBPO and SAC. For MBPO we use a re-implementation of that algorithm made available by other authors[3], which we found to give good results. Originally, the optimization of the MBPO dynamics architecture stops when the validation set sees no improvement for five epochs. To reduce computational time, we train the dynamics model for five epochs and only use the validation set to choose the elite networks. The variational dynamics uses the same dynamics architecture as MBPO, but an objective augmented with the exponential TD-error, $\exp(r_t + V(s_{t+1}) - Q(s_t, a_t))$. We clamp the TD-error between (-4,4) to avoid large gradients and numerical issues. The same actor-critic architecture is kept from SAC with two major changes: entropy term is removed from the critic optimization and the prior policy appears on the variational policy objective as regularizer. A hard-update of the prior policy to the variational policy is done after every actor optimization. Finally, we remove the log-term from the critic update (Eq. 12) as the variance in this estimator hurts the critic's convergence. For $\beta$-VMBPO, we found that including the entropy regularizer into its critic helped on the exploration of the more challenging environments.

### C.2   HYPERPARAMETERS

Table 1 lists the hyperparameters used for the OpenAI Gym benchmarks.

Table 1: Hyperparameters for OpenAI Gym benchmark

| Schedule details | |
|---|---|
| Environment steps before training | 5000 steps |
| Environment steps per epoch | 1000 steps |
| Model optimization | every 250 steps |
| Number of model rollouts | 100,000 rollouts |
| Rollout length | 1 step |
| Actor-critic updates per environment step | 20 updates |
| **Network details** | |
| Discount factor | 0.99 |
| Soft target update | 0.005 |
| Ensemble size | 7 |
| Number of elites | 5 |
| Experience buffer $\mathcal{D}_{\text{env}}$ | 1,000,000 |
| Model buffer $\mathcal{D}_{\text{model}}$ | 400,000 |
| Dynamics Network Architecture | MLP with 4 hidden layers of size 200 |
| Actor Network Architecture | MLP with 2 hidden layers of size 256 |
| Critic Network Architecture | MLP with 2 hidden layers of size 256 |
| Network optimizer | Adam |
| Non-linear layers | ReLU |
| Learning rate | 0.0003 |
| $\epsilon$ constraint | 0.1 |
| $\beta$ initialization | 100 |

### C.3   STABLE CRITIC OBJECTIVE FOR $\beta$-VMBPO

Optimizing objective (Eq. 7) can result in slower convergence for function approximators. Hence, we learn value functions w.r.t. a scaled objective resulting in the squared approximation:

$\mathbb{E}_{(s_t, a_t, s_{t+1}, r_t) \sim \mathcal{D}_{\text{model}}} \left[ \left( r_t + V_2(s_{t+1}) - Q_2(s_t, a_t) - \beta \log \frac{q(s_{t+1}|s_t, a_t)}{p(s_{t+1}|s_t, a_t)} \right)^2 \right]$. To recover the orig-

---
[3]https://github.com/Xingyu-Lin/mbpo_pytorch

inal value functions, we scale the learned value functions by $\beta$: $Q_1(s_t, a_t) = \frac{Q_2(s_t, a_t)}{\beta}$ and $V_1(s_t) = \frac{V_2(s_t)}{\beta}$.

## C.4 SETTING THE KL CONSTRAINT $\epsilon$

For environments with deterministic dynamics, the KL term should be tight at zero. In practice, this never happens when using function approximators in high-dimensional spaces. We found that using $\epsilon = 10$ was sufficient for our more simple environments. For more complicated environments, we clamp $\beta$ whenever it becomes too large (above 10,000,000).

## D  PSEUDOCODE OF $\beta$-VMBPO

This section contains the pseudocode for our algorithm, $\beta$-VMBPO.

---

**Algorithm 1** $\beta$-VMBPO

---

Initialize networks, parameters and replay buffers.
**for** each epoch **do**
  **for** each environment step **do**
    $a_t \sim \pi_\kappa(\cdot|s_t)$          ▷ Sample action from prior policy.
    $s_{t+1} \sim p(\cdot|s_t, a_t)$          ▷ Sample next state from environment.
    $\mathcal{D}_{\text{env}} \leftarrow \mathcal{D}_{\text{env}} \cup \{(s_t, a_t, s_{t+1}, r(s_t, a_t))\}$          ▷ Add tuple to experience buffer.
    **if** model optimization **then**
      $\{(s_t^i, a_t^i, s_{t+1}^i, r_t^i)\}_{i=1}^N \sim \mathcal{D}_{\text{env}}$          ▷ Sample every tuple in experience buffer.
      $\theta \leftarrow \theta - \nabla J(\theta)$          ▷ Update prior dynamics $p_\theta$.
      $\phi \leftarrow \phi - \nabla J(\phi)$          ▷ Update variational dynamics $q_\phi$.
      **for** $m = 1, 2, ..., M$ **do**          ▷ Generate rollouts using variational model.
        $s_t \sim \mathcal{D}_{\text{env}}$          ▷ Sample state from experience buffer $\mathcal{D}_{\text{env}}$.
        $a_t \sim q_\omega(\cdot|s_t)$          ▷ Sample action using variational policy.
        $s_{t+1} \sim q_\phi(\cdot|s_t, a_t)$          ▷ Sample next state using variational dynamics.
        $r_t \sim p_\theta(\cdot|s_t, a_t)$          ▷ Sample reward using prior model.
        $\mathcal{D}_{\text{model}} \leftarrow \mathcal{D}_{\text{model}} \cup \{(s_t, a_t, s_{t+1}, r_t)\}$          ▷ Add tuple to model buffer.
      **end for**
    **end if**
    **for** $k = 1, 2, ..., K$ **do**
      $\{(s_t^i, a_t^i, s_{t+1}^i, r_t^i)\}_{i=1}^B \sim \mathcal{D}_{\text{model}}$          ▷ Sample mini-batch from model buffer $\mathcal{D}_{\text{model}}$.
      $\psi \leftarrow \psi - \nabla J(\psi)$          ▷ Update critic $Q_\psi$.
      $\omega \leftarrow \omega - \nabla J(\omega)$          ▷ Update variational policy $q_\omega$.
      $\psi' \leftarrow \tau\psi + (1 - \tau)\psi'$          ▷ Update target critic $Q_\psi'$.
      $\kappa \leftarrow \omega$          ▷ Update prior policy $\pi_\kappa$.
    **end for**
    $\beta \leftarrow \beta - \nabla J(\beta)$          ▷ Update dual variable $\beta$.
  **end for**
**end for**

---

