# OpenReview forum: "Risk-Sensitive Variational Model-Based Policy Optimization"
_ICLR.cc/2024/Conference — Submitted to ICLR 2024_

### Official Review · Reviewer_dkQy · 2023-10-30

**Soundness:** 3 good
**Presentation:** 4 excellent
**Contribution:** 3 good
**Rating:** 6
**Confidence:** 3

**Summary:**

This paper addresses the issue of risk-seeking policies in model-based RL-as-inference which arises due to learning optimistic dynamics.
The paper highlights how optimistic dynamics lead to risk-seeking policies in Figure 1.
The authors then introduce a risk parameter ($\beta$) to the RL-as-inference objective that enables (nonlinear) interpolation
between optimistic dynamics (for low $\beta$) and the true dynamics (for large $\beta$).
They then highlight connections between their ELBO (objective) and the risk-sensitive exponentiated utility for SafeRL.
Hence the paper's name, I guess.
Drawing on connections between this objective and constrained optimization with Lagrange multipliers,
they propose a method to automatically adapt $\beta$.
Intuitively, their method restricts learning such that the KL divergence between the variational and prior dynamics remains
below a threshold $\epsilon$.
That is, the variational dynamics can be optimistic as long as they're close to the real dynamics.
They provide a theoretical analysis of their risk-sensitive method and
demonstrate that it overcomes issues of risk-seeking behaviour in a stochastic tabular environment.
They also show that it fixes issues with VMBPO in deterministic continuous environments (Mountain Car), which I found to be a surprising (and interesting) result.
Finally, they show that it scales to high(ish) dimensional benchmarks such as Hopper/Walker2D/HalfCheetah.

**Strengths:**

Overall, I think this is a good paper.
It highlights a known (yet understudied) problem in the RL-as-inference framework (that of optimistic dynamics)
and presents a novel solution to combat it.
I found the paper well-written, I particularly liked Sec. 2 and Figure 1 as it was helpful for providing intuition for
the issue that the paper is trying to solve.
The theoretical analysis seems correct, although I have not checked it in detail.
The experiments are also well-structured and easy to follow.

**Weaknesses:**

I do not have any major weaknesses with this paper.
Nevertheless, I will provide some comments with the aim of helping the authors further improve their manuscript.

The dynamics models are learned using an ensemble of MLPs which parameterize Gaussian densities.
That is, each ensemble member learns a heteroscedastic noise model which captures the MDP's transition noise.
Although not detailed in the paper, I assume the ensemble is a uniformly weighted mixture
$q_{\phi}(s_{t+1}\mid s_t,a_t) = \frac{1}{B} \sum_{b} q_{\phi_{b}}(s_{t+1}\mid s_t,a_t)$
where each member is given by,
$q_{\phi_{b}}(s_{t+1}\mid s_t,a_t) = \mathcal{N}(\mu_{\phi_{b}}(s_{t},a_{t}), \Sigma^{2}(s_{t}, a_{t}))$.
The ensemble then captures the epistemic uncertainty arising from limited data.
How do you calculate the KL divergence between the variational and prior dynamics given that they are Gaussian mixtures?
Do you assume that the output is unimodal and fit a single Gaussian to the mixture?
Subtleties like this could impact the performance of VMBPO and $\beta$-VMBPO so they should be detailed in the paper.

Also, how do you train the ensemble?
Is each member's parameters initialised differently? Or is each member trained on different data?

In my mind, the notion of risk applies in uncertain environments.
In Fig. 4b, how is this dynamics model risky?
Isn't it just optimistic and stuck in a local optimum induced by the optimism?
I'm happy to be corrected here.

I'm also wondering if Fig. 4 used an ensemble of MLPs without probabilistic heads?
The paper states that all experiments use probabilistic heads.
But this environment is deterministic so this feels like an odd thing to do.

How do you populate $\mathcal{D}\_{\text{model}}$? I mainly wanted to know how the initial state is sampled.
Do you sample randomly from the state space or do you sample a state from the replay buffer and then perform a rollout?
I later found this in Alg. 1 in the appendix. Perhaps reference this algorithm from paragraph 2 of Sec. 3.3 so that it's clear at this point how $\mathcal{D}_{\text{model}}$ is populated.

Minor things:
- There is an abuse of notation which should be made clear. That is, $Q(s_t,a_t)$ in Eq. 2/3 is different to Eq. 5. It could be made $Q(s_t, a_t;\beta)$ in Eq. 5 or there could be a sentence to explain that the notation is being abused.
- The first paragraph of Sec. 3 should reference Appendix D and not just the appendix.
- How is $V_{\psi}'(s_t)$ different to $V_{\psi}(s_t)$? I couldn't find this anywhere in the text.
- Third paragraph of Sec. 3.3 should have $\pi_{\kappa}(a_t\mid s_t)$ instead of $\pi(a_t\mid s_t)$?
- Theorem 4.3 uses "Thm 4.1" and "Thm 4.2" but the rest of the paper uses "Theorem 4.1". Be consistent.
- This sentence in the related work feels out of place. "Variational approximate RL-as-inference by maximizing a lower bound on the marginal likelihood.". Is part of the sentence missing?
- Fig. 2 caption. "Learn" should be "Learned" or "Learnt".
- In Fig. 2 it's not clear the grey area is a cliff. I'd suggest overlaying text saying "CLIFF" on the grey area.
- Fig. 3 specifies $\beta$ as a parameter. It might be clearer to use $\beta^0$ to indicate that this is the initial value of $\beta$. At the moment it gives the impression that $\beta$ is fixed.
- Fig. 3 uses "Beta-VMBPO" and Fig. 5 uses "Beta_VMBPO". Be consistent.
- Appendix C.1 "Pytorch" should be "PyTorch".
- Fig 2/3/4/5 axis numbers and labels are too small.
- Fig 3/5 legend is too small.
- In Fig. 5 the left/middle plots don't show the curves converging. Do they converge to the same value as SAC? It feels suspect not to show $\beta$-VMBPO not converging to the same value as SAC.

**Questions:**

- Have you added more details on the ensemble of probabilistic MLPS dynamics models?
- Have you addressed my issue with Fig. 5? That you don't show $\beta$-VMBPO converging to the same value as SAC.

---

> ### Author Response · Authors · 2023-11-20
>
> **KL divergence calculation**
>
> Our approach computes the KL divergence between two probabilistic networks --- the network with best validation loss for each ensemble model. We then estimate KL divergence using Monte Carlo integration where the samples are generated using the variational probabilistic network. We agree with the reviewer that this choice should be detailed in the paper and we will include a brief discussion.
>
> **Ensemble optimization**
>
> Our ensemble is composed of seven networks, each with a different random initialization. Component networks are optimized on the same data and rollouts are generated by randomly selecting a network out of the five networks with smallest the validation loss.
>
> **Optimistic vs risk-seeking behavior**
>
> We agree with the reviewer that in this example the dynamics are acting optimistic rather than risk-seeking. This distinction applies in general to the framework even when the environment is uncertain --- the policies are optimistic (risk-ignoring) rather than risk-seeking. Nonetheless "risk-seeking" is the more common parlance in the literature.
>
> **Probabilistic model for deterministic environment**
>
> A probabilistic model offers some benefits even when the environment dynamics are deterministic. We find that a probabilistic approach provides robustness to poor NN function approximation during early stages of learning.  In our setting, it also plays an important role in smoothing the KL constraint --- under deterministic dynamics the KL is undefined everywhere except on the value $q=p$.
>
> **Population of the model buffer $\mathcal{D}_{model}$**
>
> Yes, we sample a state from $\mathcal{D}_{env}$ and then produce a roll-out using our variational policy and dynamics. We will include this clarification in the main text.
>
> **Performance convergence**
>
> We do not provide guarantees that our method will converge to the same value as SAC. Model-based RL methods are known to perform worse asymptotically than model-free methods. We did not train the models to convergence as it becomes computationally prohibitive. Instead, we use the standard number of training iterations in the model-based RL literature for these environments that is matched to comparisons in the existing literature (e.g. MBPO).

---

> > ### Comment · Reviewer_dkQy · 2023-11-22
> >
> > Thank you for your response. I have read the author's comments, the other reviewers' comments and the public comments.
> >
> > Whilst I found the paper clear, with regards to the definitions of optimism/risk used in the paper, I have two issues that have made me decrease my score (from 8 to 6), and my confidence (from 4 to 3).
> >
> > First (as highlighted by Reviewer wKFul), the inconsistency of the MBPO experiments relative to the original MBPO paper should be addressed. I recommend the authors rename the current MBPO results to make it clear how they differ from MBPO. Then I'd suggest adding results for a good implementation of MBPO. Whilst both results have value, it is misleading to report MBPO results for a modified version of MBPO.
> >
> > Second, I was not aware of papers highlighted by the public comment. These seem very relevant and should at least be cited in the related work.

---

> ### Author Response · Authors · 2023-11-22
>
> We agree with the reviewer that this difference should be made more explicit in the experiment section. We have uploaded a revised version that addresses this modification in the introduction of the baselines algorithms and renames the MBPO performance curve to MBPO (Fixed iterations).
>
> We agree that these references are very relevant, and we will include them in the Related Work section.

---

### Official Review · Reviewer_9Tpq · 2023-11-01

**Soundness:** 3 good
**Presentation:** 3 good
**Contribution:** 3 good
**Rating:** 6
**Confidence:** 3

**Summary:**

The study introduces a risk-sensitive approach to reinforcement learning (RL) by framing RL as Bayesian inference within a probabilistic graphical model, termed "control as probabilistic inference." Conventional model-based control-as-inference does not consider risk-sensitive policy learning. The authors introduce the β-VMBPO algorithm, a risk-sensitive variant of the variational model-based policy optimization (VMBPO) algorithm. The novelty of this method lies in its dual descent optimization, which incorporates the risk parameter β as an adaptive parameter. This approach is bolstered by a thorough theoretical analysis. Empirical evaluations validate the efficacy of this risk-sensitive methodology, demonstrating its advantage over traditional methods.

**Strengths:**

1. The paper establishes a clear connection between dual-descent and risk-sensitive variational inference, enabling the optimization of \beta in a logical manner.
1. Comprehensive theoretical analyses are provided.

**Weaknesses:**

1. The research's novelty appears limited. Previous studies have already introduced \beta. If the paper's main contribution centers around a reinterpretation of \beta through the lens of dual-descent, there's a need for a clearer distinction between this work and earlier research.
1. Although the theoretical approach is interestingly anchored in the RL-as-inference framework, the paper falls short in its discussion regarding the interpretation of the parameter \beta within the context of variational inference. A more in-depth exploration and dialogue from the variational inference standpoint would significantly enrich the paper.

**Questions:**

1. In the "PRELIMINARIES" section, Equation (5) appears to divide the reward by \beta directly. While I recognize that this might not be the primary focus of the authors, readers could wonder how such a minimal change can imbue the method with risk sensitivity. A brief and intuitive explanation addressing this would be valuable.
1. To my understanding, the pioneering work introducing variational inference to model-based reinforcement learning is "VI-MPC" by Okada et al. (2020). While the goals of the two studies might differ, I believe it's pertinent for the authors to reference this foundational work.
Reference:
Okada, Masashi, and Tadahiro Taniguchi. "Variational inference MPC for Bayesian model-based reinforcement learning." Conference on Robot Learning. PMLR, 2020.
1. In Figure 1, an overlaid directed graph makes the illustration less intuitive. The representation seems muddled, especially when juxtaposed with the legends. It's evident that this figure could benefit from some revisions for clarity.
1. Dual optimization emerges as a crucial component of this research. Given that several potential readers may be unfamiliar with this concept, adding a reference, such as a standard textbook on convex optimization or a pertinent tutorial paper, would be of great assistance.

---

> ### Author Response · Authors · 2023-11-20
>
> **Risk parameter novelty**
>
> Although the risk parameter $\beta$ has been considered in previous work, its role has been understudied---as reviewer dkQy correctly points out. The need for, and impact of, practical tuning of the risk parameter is further supported by the public comment on this paper and associated work (see Brendan O'Donoghue's comment above).  In the safe RL literature (Garcia \& Fernandez, 2015; Shen et al.~2013; Noorani \& Baras, 2022) it has been shown that $\beta > 0$ exhibits risk seeking behavior and recovers the risk-neutral objective when $\beta \to 0$, but no method has been proposed to adequately tune this parameter. In VMBPO (Chow et al, 2021), it was demonstrated that learning degradation might occur due to variance amplification, and this parameter can alleviate this issue, but tuning it was not addressed. Our work, studies how $\beta$ affects the posterior dynamics and the optimal policy, and demonstrates that it is a key component to obtain policies that can be applied to the real environment.
>
> **$\beta$ within the context of variational inference**
>
> In the variational lower bound, the parameter $\beta$ limits the disagreement between the variational dynamics and the true dynamics through the KL penalty. Fig. 1b shows this behavior where we found the optimal posterior dynamics for different $\beta$ values. $\beta$-VAE (Higgins et al, 2017) studied a similar trade-off, albeit in a different context where $\beta$ limits the capacity of the variational distribution thus permitting the method to learn disentangled representations. In our work, the constraint is w.r.t.~the environment dynamics producing less risk seeking variational distributions.  We will include this discussion in the Related Work section.
>
> Reference:
>
> Higgins, Irina, et al. "beta-vae: Learning basic visual concepts with a constrained variational framework." International conference on learning representations. 2016.
>
> **Shared reference VI-MPC**
>
> We appreciate the shared reference by the reviewer. We agree that it is relevant and we will include it accordingly in our work. But we want to differentiate this method from variational model-based RL algorithms. The variational inference for VI-MPC only affects the policy --- similarly to SAC (Haarnoja et al, 2018) and MPO (Abdolmaleki et al, 2018). In VI-MPC model dynamics do not appear as part of the optimization objective, contrary to VMBPO and our work where the model is part of the variational optimization.
>
> **Dual optimization reference**
>
> We agree with the reviewer that the reader may benefit from including a reference for dual optimization. We will include it in our work.

---

> ### Comment · Reviewer_9Tpq · 2023-11-21
> **Thanks**
>
> Thanks for your clarification.
>
> >In Figure 1, an overlaid directed graph makes the illustration less intuitive.
>
> I hope the authors deal with this problem in their revised version as well.
>
> > In VI-MPC model dynamics do not appear as part of the optimization objective, contrary to VMBPO and our work where the model is part of the variational optimization.
>
> Agree and understood.
> This comment reminded me the following paper as well. This is just for your information.
>
> Okada et al. "Planet of the Bayesians: Reconsidering and improving deep planning network by incorporating bayesian inference." 2020 IEEE/RSJ International Conference on Intelligent Robots and Systems (IROS). 2020.
>
> The score has been updated based on your feedback.

---

### Official Review · Reviewer_wKFu · 2023-11-02

**Soundness:** 2 fair
**Presentation:** 3 good
**Contribution:** 2 fair
**Rating:** 3
**Confidence:** 4

**Summary:**

Briefly, the original VMBPO algorithm learns a dynamics model
where the dynamics transitions are weighted according to the
exponential of the value of the transitions, in accordance
with the control as inference approach.
This paper modifies the VMBPO algorithm to include an inverse
temperature parameter $\beta$ so that the risk seeking behavior of
VMBPO could be modulated. Another change is the way how the likelihood
ratio $q/p$ of the variational model to the true dynamics is computed:
in the original VMBPO, they use a direct estimator for the ratio,
whereas in the current work, they learn two models, one for $q$ and
one for $p$. Moreover, they suggest to tune the $\beta$ parameter by
setting a target KL divergence, and optimizing $\beta$, to achieve the
target in a similar way how the SAC algorithm optimizes their inverse
temperature parameter to achieve a target entropy.

The experiments included two simple tabular domains to show how
modulating $\beta$ controls the risk seeking behavior of the method,
and simple control tasks: Mountain Car, Half Cheetah, Walker2D, Hopper
(note that the abstract says they tried DeepMind Control Suite, but this
seems inaccurate, it seems the work only contains OpenAI Gym experiments).

**Strengths:**

1. The necessity of $\beta$ in the VMBPO algorithm is clear, so the
approach is well-motivated from this point of view.
2. There was a study looking at how the tuning of the $\beta$ works
in an MDP domain that showed that the method is robust to initializations
in this domain suggesting reliability of the method.

**Weaknesses:**

1. While the necessity of the $\beta$ term is clear, why one
should use a VMBPO-based approach to begin with was not clear to me.
The majority of the discussion was about how VMBPO can be risk seeking,
and how modulating the $\beta$ term can prevent the risk seeking
behavior, but one can also avoid the risk seeking behavior by simply
not using VMBPO. I think the advantage of using the control as inference
approach should also be explained and demonstrated.

2. There were only 3 OpenAI Gym benchmark environments. It would have been
good to include more, e.g., Ant and Humanoid.

3. The experimental results in HalfCheetah and Walker2d do not match the
published results in the original MBPO paper
(https://arxiv.org/pdf/1906.08253.pdf, figure 2). In particular, HalfCheetah
seems to go clearly over 10000 (reaching around 12000), whereas it barely
reaches 10000 in this paper, also in Walker, the results go clearly
above 4000, but barely reach 4000 in this paper. The results on Hopper
were indistinguishable between MBPO and Beta_VMBPO. Also the improvement
is quite marginal. I am not confident the method reliably improves over
MBPO. Also, the number of random trials was 5, which is low.

4. While there were experiments on smaller tasks aimed at explaining how the
method works, the only statistics provided in the Gym tasks were the reward
curves. It would have been better to provide some other statistics that
demonstrate that the risk sensitivity control is working as intended.

5. The computational time was not discussed. In particular, as you compute
two models, $p$ and $q$ does that affect the computational time?

6. I think the risk seeking behavior of VMBPO and that $\beta$ modulates
this are obvious, and I think too much space was used for explaining
these points. I think the simple experiments on the tabular
domains and on the mountain car were redundant, as they do not show any non-obvious
result. For example, in the mountain car task, if the $\beta$ parameter is set
sufficiently large, then clearly the variational model should try to match the
true dynamics, and the method should work. I would have liked to see a result
that requires tuning $\beta$ and cannot be achieved without simply trying to
learn the true model.

7. Equation 8 suggests an inequality constraint for the KL, yet in
Equation 9, you are optimizing for an equality constraint (as is done
in SAC, also there was no reference to SAC). I was not completely
convinced with this.

8. The works lists the necessity to tune the KL constraint target
$\epsilon$ parameter as a limitation. However, it is also necessary to
set an initial $\beta$ value, as well as a learning rate for
$\beta$. While the experiments looked at the sensitivity to these
parameters in a simple tabular task, the senstivity was not examined
in the other tasks, so it is not completely clear how reliable the
method is.

9. VMBPO and the suggested Beta_VMBPO have more differences than simply adding in a beta value. I think there should have been ablation studies on the different components of Beta_VMBPO, e.g., the tuning of the beta value, etc. (performed in the OpenAI Gym tasks)

**Questions:**

In this work, the ratio q/p is estimated by learning two models:
one for q and one for p, then taking the ratio. In the origianl VMBPO,
they use a direct estimator, v, for q/p. Vapnik's principle suggests
that learning a direct estimator is typically better than solving
more general intermediate steps. Have you compared with this method,
and why did you choose to learn two models?

Please also feel free to respond to any of the listed weaknesses.

Bottom of page 12, there's a typo: "p(.|s_t, s_t)"

---

> ### Author Response · Authors · 2023-11-20
>
> We appreciate the multiple points raised by the reviewer. We have incorporated an extra comparison for OpenAI benchmarks (Ant) and an ablation study for the initialization and learning rate of beta in the appendix. We now address some points made by the reviewer.
>
> **Motivation for control as inference**
>
> The risk seeking behavior is not just an issue with VMBPO, but a fundamental problem with the control-as-inference framework that merits further investigation. Despite this behavior the methodology has inspired many novel techniques and algorithms with good performance in high dimensional spaces (see Related Work). These methods, such as MaxEnt RL, avoid problematic risk-seeking behavior by imposing additional constraints on the variational approximation that lead to weak bounds on the marginal likelihood. Our work demonstrates that these constraints are unnecessary, allowing for more flexible variational approximations.  By further emphasizing the connection to the exponential utility we establish $\beta$-VMBPO as an instance of Risk-Sensitive RL for which there is extensive benefit in practical applications (Garcia and Fernandez, 2015). This work opens a path for unconstrained variational methods in the control-as-inference framework.
>
> **Experimental results gap**
>
> Implementation details are documented in Appendix C.1.  To reduce computation and obtain a fair comparison we impose consistent optimization of the dynamics architecture across all methods.  The original MBPO optimization stops when validation loss sees no improvement for five consecutive epochs, leading to long computation times. We instead optimize the dynamics network for a fixed number of epochs (See appendix C.1). All methods use the exact same number of iterations and same network architectures making the comparison consistent across algorithms. The number of random trials (five) is the same as was used in the original MBPO paper.
>
> **"Inequality constraint for the KL..."**
>
> Our optimization corresponds to an inequality constraint $KL(q||p) \leq \epsilon$.  This results in the Langrangian $\mathbb{E}_q[\sum r(s_t, a_t)] -\beta[KL(q||p)-\epsilon]$. The dual optimization results in the loss from Eq. (9) which is correct by complementary slackness and the necessary KKT conditions.  SAC does not optimize an equality constraint, but an inequality constraint just as our method. We make reference to SAC after Eq. (8) to compare the model-free approach where this term the variational dynamics is removed.  We also discuss the relationship with SAC in the related work section where we compare the similarities between our dual optimization and SAC's optimization, with the major difference being risk modulation instead of controlling the policy's entropy.
>
> **"The ratio q/p is estimated by learning two models..."**
>
> Through extensive experimentation we found that directly estimating the log ratio using the convex conjugate of Chow et al. (2021) was unstable for environments with deterministic dynamics, such as the OpenAI Gym benchmarks. Our estimator solves this problem by relaxing the assumption of determinism with two dynamics models represented by network ensembles.

---

> ### Comment · Reviewer_wKFu · 2023-11-22
>
> Thanks for the comment, but unfortunately I have not changed my mind.
>
> Regarding the constraint: (EDIT: removed comment on constraint as I may have misunderstood something). However, I am still of the opinion that the discussion of SAC is not sufficiently clear. The method for tuning the parameter is basically exactly the same. I believe this should be made clear, e.g., by equation 9. To me, the discussion of SAC was too obfuscated.
>
> The performance you showed on Ant also does not match previous results.
>
> My impression is that any gains you are seeing in the experimental results are due to implementation changes rather than a fundamental advantage of your method compared to MBPO (i.e., the settings you chose may leave an impression that your method performs better, but with other settings, MBPO would be better or the same). For me to change my assessment, I would request that the method is compared against a well-performing implementation of MBPO, or that you show a clear win on some task. Computation times and statistics other than reward curves were still not discussed. Regarding only using 5 seeds, this was still accepted at the time of the MBPO paper, but it has received a lot of criticism since then, e.g., see "Deep Reinforcement Learning at the Edge of the Statistical Precipice" (https://arxiv.org/abs/2108.13264) for a discussion, and better assessment methods.
>
> Currently, the main clear result in the paper seems to be the necessity for the $\beta$ parameter, but I believe that this alone is not sufficiently significant. The public commenter has also kindly pointed out many related works that also use a $\beta$ parameter. Also, the use of an ensemble for aleatoric uncertainty does not seem well founded. So, I'm afraid I will not be changing my assessment based on the comments in the rebuttal. I would encourage the authors to revise the paper, and try running experiments when comparing against a well-performing version of MBPO.

---

> > ### Author Response · Authors · 2023-11-22
> >
> > **SAC discussion:**
> >
> > We respectfully disagree with the statement that these procedures are exactly the same. Both methods are similar in their optimization of a dual parameter which is a standard technique to deal with an inequality constraint. This is where the similarities between these two methods end as the dual optimization in SAC computes an estimate of its entropy using its actor (policy) while our method uses its two dynamics models (variational and prior) to compute an estimate of their KL divergence. The role of their dual parameters is also widely distinct. SAC affects the entropy of its policy, while our method affects its variational dynamics.
> >
> > **$\beta$ parameter**
> >
> > The work cited by the public commentator also considers the optimization of a parameter $\beta$. However, this is done in the context of K-learning where the parameter controls a trade-off of epistemic uncertainty and expected return. In contrast, our parameter balances expectation and variance of the return by constraining the divergence between variational and prior dynamics. This fundamental difference results in very different optimization objectives and approaches --- in the former the parameter is optimized to reduce Bayes regret, while in the latter the parameter is optimized to modulate the behavior of the variational dynamics.
> >
> >
> > **Ensemble model:**
> >
> > In our work, we don't make the claim that the ensemble of neural networks was chosen to combat aleatoric uncertainty. Instead, we use it because of its competitive performance w.r.t. model-free algorithms and its robustness against policy exploitation (Chua et al, 2018; Janner et al. 2019).

---

> > > ### Comment · Reviewer_wKFu · 2023-11-23
> > >
> > > **SAC discussion:**
> > >
> > > The method for tuning the dual parameter is the same. The only difference is that in SAC they take the difference between the entropy and target entropy, whereas in your paper, you use the difference between the KL and target KL. The method for tuning the $\beta$ parameter was claimed as a main contribution of the paper, but unfortunately I do not consider it very novel (moreover, I would request that the relationship to the method in SAC is made clear). This is not a major issue by itself, but I would want to be convinced by the experiments that the method is working well. Moreover, did you have any experiments with fixed $\beta$ values? It would be necessary for me to show that the tuning clearly beats using a fixed $\beta$ value.
> > >
> > > **Remaining perceived issues in the paper:**
> > >
> > > The biggest issue for me is that the experiments do not reproduce previous performance (and the results you show for your algorithm are also not competitive with current methods). I understand that MBPO is slow in its original formulation, but if your method really works, then it should not be too difficult to run your algorithm in a similar formulation. If the method is effective, I would expect it to give better performance with the parameters that you have already tuned. For me to change my assessment of the work, running these experiments would be necessary.
> > >
> > > Some additional concerns I have are:
> > > For the new ablation study on the learning rate and initial $\beta$ parameter, why did you choose Hopper? This was the only environment where there was clearly no advantage to your method compared to other algorithms, so perhaps this environment was not very sensitive to tuning. Also, the method was not completely robust to the setting of the initial $\beta$ value: the performance drops considerably for $\beta=10$ (it is particularly worrying that 1, and 100 seem to give reasonable performance, so we can't identify a clear trend of when the performance drops). Considering that you originally only had 3 environments, it may be better to show the results for all of the environments. Also, how does the performance differ when the $\epsilon$ parameter is changed? I didn't find a study on this. The setting of $\epsilon$ was also not clear to me. In Table 1, it is written that $\epsilon=0.1$, but in section C.4, it is written that $\epsilon=10$.
> > >
> > > You wrote: "Through extensive experimentation we found that directly estimating the log ratio using the convex conjugate of Chow et al. (2021) was unstable for environments with deterministic dynamics, such as the OpenAI Gym benchmarks."
> > > Please provide the evidence for this in the article, as it could be an additional contribution of the work.
> > >
> > > Regarding the use of ensemble models, I guess your explanation is that you used an ensemble to improve the performance, but considered aleatoric uncertainty for the KL computation. In your response to reviewer dkQy you stated that you will include an explanation of the KL computation, but I didn't find this in the paper yet. How exactly you did this was also not completely clear to me.
> > >
> > > There are also many points in my review that were never addressed in the rebuttal, e.g., what is the computation time? Do you have any additional statistics (e.g., how does the KL change during learning, how does $\beta$ change during learning in the MuJoCo tasks)? I would request more extensive experimental results in the paper for me to change my assessment. Currently, the results are not convincing to me to show that the method is indeed working well and providing an advantage over a regular MBPO based approach. I do not expect myself to change my score during this submission phase, but if the authors wish to continue the rebuttal, then please address all of the points in my reviews, either agreeing or disagreeing.

---

### Public Comment · ~Brendan_O'Donoghue1 · 2023-11-13
**Clarity on uncertainty and 'RL as inference'**

The title of this paper piqued my interest, since I have done a lot of work along very similar lines.  This is a nice paper, but I think that there are some issues of clarity and it might help to position the paper better with existing literature on this topic, which I do not think is done correctly at the moment.

The first thing that jumps out at me is that the paper does not clearly delineate between epistemic and aleatoric uncertainty. It seems to me like the paper is focusing on aleatoric, but it’s difficult to fully understand and is confused further by the use of an ensemble of neural networks which would typically be used to estimate epistemic uncertainty. For instance, in the abstract the sentence “it is known that model-based RL-as-inference learns optimistic dynamics and risk-seeking policies that can exhibit catastrophic behavior” conflicts with the fact that ‘optimism in the face of uncertainty’ is the primary heuristic used to learn optimal policies efficiently. In other words, exploration is inherently epistemic-risk-seeking. I can only assume the authors here are referring to aleatoric uncertainty, but that needs to be clarified. I think some of the confusion here might arise from the previously published ‘RL as inference’ framework, which uses confusing terminology and has some serious weaknesses (see refs listed below). The sentence ‘This so-called RL-as-inference framework facilitates a principled solution to the exploration-exploitation trade-off by adapting the policy to posterior uncertainty’ is unfortunately not true, as it does not account for epistemic uncertainty at all.

In regards to the literature, I have written several papers on variational and risk-sensitive approaches to RL, which I think are very relevant to this paper and the authors should be aware of them and position their paper accordingly:

- In ‘Variational Bayesian Reinforcement Learning with Regret Bounds‘ (https://arxiv.org/abs/1807.09647) we derive a risk-seeking variational model-based RL algorithm (K-learning) that has a similar flavor to $\beta$-VMBPO, in particular the use of a risk-sensitive exponential utility function and tuning the $\beta$ parameter for performance ($\beta$ is equivalent to $\tau$ in that paper). In that paper we show that the $\beta$ term (ie, $\tau$) can be optimized jointly with the value functions. This paper is explicitly about efficient RL (aka, exploration) and therefore primarily concerned with epistemic uncertainty.

- In ‘Making Sense of Reinforcement Learning and Probabilistic Inference’ (https://arxiv.org/abs/2001.00805) we highlight that the ‘RL as inference’ approach epitomized by the survey paper by Levine is deeply flawed, and propose the previously mentioned K-learning and Thompson sampling algorithms as alternatives.

- In ‘Efficient Exploration via Epistemic-Risk-Seeking Policy Optimization’ (https://arxiv.org/abs/2302.09339) we convert the risk-seeking model-based algorithm over value functions, to a risk-seeking model-free algorithm over policies, where again the temperature $\tau$ (ie, $\beta$) is learned on the fly jointly with the policy via a saddle-point problem. Ultimately applying an actor-critic style algorithm to the objective.

- Having a single temperature ($\tau$ or $\beta$) term is clearly sub-optimal - imagine the case where one entire region of the state-space has zero uncertainty and another region has high uncertainty. In the zero uncertainty area the agent should be (mostly) greedy, so $\beta$ ($\tau$) equal to zero is optimal there, but in the other region the agent should use a non-zero temperature. In ‘Variational Bayesian Optimistic Sampling’ (https://arxiv.org/abs/2110.15688) we managed to break apart the single scalar temperature ($\beta$) into one that is action dependent via a refined analysis, showing better performance.

- In ‘Probabilistic Inference in Reinforcement Learning Done Right’ (https://openreview.net/forum?id=_0ggHCj8cPx, to appear at Neurips this year) we generalized the above paper to the full MDP case, that is, an algorithm with a temperature ($\beta$) that is state and action dependent rather than a scalar. Moreover, we tied it all back to a principled variational approach to the ‘RL as inference’ framework that fixes its major issues.

This is a lot of papers to be throwing at you, and I apologize for that, but I and others have been working on exactly this area for quite some time and making good progress, in particular in the area of variational, risk-sensitive policies where the temperature parameter ($\beta$) is learned. After the review period is over and anonymity is no longer required I would be happy to chat with the authors of this paper about connections and possible future work in more detail if they wish.

---

> ### Author Response · Authors · 2023-11-21
>
> We appreciate your interest and comments of our work. We agree that these references are relevant, and we will incorporate them in the Related Work section.  In particular, we are familiar with the paper "Making Sense of Reinforcement Learning", but due to an oversight it was not included in the submitted draft.  We now clarify some of your raised questions:
>
> **Aleatoric vs epistemic uncertainty**
>
> We position our paper w.r.t.~the RL-as-inference framework presented in Levine (2018).  In keeping with that literature we do not explicitly distinguish between epistemic and aleatoric uncertainty.  This framework optimizes a Risk-sensitive RL objective (the exponential utility) that simultaneously optimizes a trade-off between the expectation and variance of the return.  As a result, our method incorporates aleatoric uncertainty into its value-functions.  We choose to represent the dynamics model with an ensemble of neural networks because of their competitive results w.r.t. model-free algorithms (Chua et al, 2018).  We agree that explicit modeling of, both, aleatoric and epistemic uncertainty is an important future direction in this space.
>
> **Optimistic dynamics vs Optimism in the face of uncertainty**
>
> We clarify that the usage of "optimism" is different in these two contexts. Optimism in the face of uncertainty provides a simple solution to the exploration-exploitation trade-off --- an optimistic action is either a known good action (exploitation) or its outcome is unknown (exploration) which updates the prior belief for this action. In contrast, the optimism in model-based RL-as-inference comes from the aleatoric uncertainty which won't be reduced as the agent continues exploring. This is a key difference to the epistemic uncertainty component in K-learning which reduces as state-action pairs are explored. Nevertheless, we demonstrate that the risk parameter can modulate this aleatoric uncertainty allowing the agent to explore the environment at the beginning and obtain an optimal policy at the later stages of learning.

---

> > ### Public Comment · ~Brendan_O'Donoghue1 · 2023-11-21
> > **Thanks**
> >
> > Great, thanks for clarifying!
> >
> > I would suggest that you are very explicit in the paper that you are dealing with aleatoric uncertainty only (eg, safe RL), just to help the reader understand because it may be confusing otherwise. Also, ensembles are typically used to estimate epistemic uncertainty, so I think some justification for their use in your setup is required, again to help the reader understand. Similarly 'optimism' is a very loaded term in RL, so if you want to use it to refer to aleatoric randomness artificially inflating the value function, then I would recommend either using a different term, or being very explicit about how you are defining it.
> >
> > Finally, the way that people typically estimate the aleatoric uncertainty nowadays is via distributional RL (ie, learn the entire distribution of possible returns, rather than just a point estimate), so it is worth positioning this paper with respect to that body of work too.

---

### Author Response · Authors · 2023-11-20

We appreciate the constructive feedback of the reviewers and the time they have invested in these reviews. We have provided individual responses to each reviewer below. For brevity we do not address minor typos, but they will be corrected in the final version.

---

### Meta-Review · Area_Chair_iDTP · 2023-12-14

**Metareview:**

The paper presents a new algorithm to adaptively adjust policy risk based on environment dynamics for risk-sensitive (safe) RL, leading to improved performance in various RL tasks. The paper evaluates the effectiveness of the proposed algorithm through experiments on both tabular and DeepMind Control Suite simulations. The results show improvements in modulating optimism in posterior dynamics, with limited comparisons.

The paper's contribution will be more solid by providing a comprehensive evaluation of the proposed approach and practical implications in real-world RL applications.

The public comment by Dr. Brendan O'Donoghue is helpful to clarify the scope of the research. This is a borderline paper, the authors should make the clarifications and improve the paper.

**Justification For Why Not Higher Score:**

The authors should make the clarifications and improve the paper as suggested.

**Justification For Why Not Lower Score:**

N/A

---

### Decision · Program_Chairs · 2024-01-16

Reject